# Polygenic Score for Body Mass Index Is Associated with Disordered Eating in a General Population Cohort

**DOI:** 10.3390/jcm9041187

**Published:** 2020-04-21

**Authors:** Mohamed Abdulkadir, Moritz Herle, Bianca L. De Stavola, Christopher Hübel, Diana L. Santos Ferreira, Ruth J. F. Loos, Rachel Bryant-Waugh, Cynthia M. Bulik, Nadia Micali

**Affiliations:** 1Department of Psychiatry, Faculty of Medicine, University of Geneva, CH–1205 Geneva, Switzerland; mohamed.abdulkadir@unige.ch; 2Great Ormond Street Institute of Child Health, University College London, London WC1N 1EH, UK; moritz.herle.12@ucl.ac.uk (M.H.); b.destavola@ucl.ac.uk (B.L.D.S.); 3Department of Biostatistics & Health Informatics, Institute of Psychiatry, Psychology & Neuroscience, King’s College London, London SE5 8AB, UK; 4Social, Genetic & Developmental Psychiatry Centre, Institute of Psychiatry, Psychology & Neuroscience, King’s College London, London SE5 8AF, UK; christopher.1.huebel@kcl.ac.uk; 5UK National Institute for Health Research (NIHR) Biomedical Research Centre, South London and Maudsley Hospital, London SE5 8AF, UK; 6Department of Medical Epidemiology and Biostatistics, Karolinska Institutet, SE-171 77 Stockholm, Sweden; cynthia_bulik@med.unc.edu; 7Medical Research Council Integrative Epidemiology, University of Bristol, Bristol BS8 2BN, UK; diana.santosferreira@bristol.ac.uk; 8Population Health Sciences, Bristol Medical School, University of Bristol, Bristol BS8 2PS, UK; 9The Charles Bronfman Institute for Personalized Medicine, The Mindich Child Health and Development Institute, Icahn School of Medicine at Mount Sinai, New York, NY 10029, USA; ruth.loos@mssm.edu; 10Maudsley Centre for Child and Adolescent Eating Disorders, Michael Rutter Centre for Children and Young People, Maudsley Hospital, London SE5 8AZ, UK; rachel.bryant-waugh@slam.nhs.uk; 11Department of Psychiatry, University of North Carolina at Chapel Hill, Chapel Hill, NC 27599, USA; 12Department of Nutrition, University of North Carolina at Chapel Hill, Chapel Hill, NC 27599, USA; 13Department of Pediatrics, Gynecology and Obstetrics, University of Geneva, CH–1205 Geneva, Switzerland

**Keywords:** Avon Longitudinal Study of Parents and Children (ALSPAC), body mass index, disordered eating behaviors, disordered eating cognitions, polygenic scores

## Abstract

Background: Disordered eating (DE) is common and is associated with body mass index (BMI). We investigated whether genetic variants for BMI were associated with DE. *Methods*: BMI polygenic scores (PGS) were calculated for participants of the Avon Longitudinal Study of Parents and Children (ALSPAC; *N* = 8654) and their association with DE tested. Data on DE behaviors (e.g., binge eating and compensatory behaviors) were collected at ages 14, 16, 18 years, and DE cognitions (e.g., body dissatisfaction) at 14 years. Mediation analyses determined whether BMI mediated the association between the BMI-PGS and DE. *Results*: The BMI-PGS was positively associated with fasting (OR = 1.42, 95% CI = 1.25, 1.61), binge eating (OR = 1.28, 95% CI = 1.12, 1.46), purging (OR = 1.20, 95% CI = 1.02, 1.42), body dissatisfaction (Beta = 0.99, 95% CI = 0.77, 1.22), restrained eating (Beta = 0.14, 95% CI = 0.10, 1.17), emotional eating (Beta = 0.21, 95% CI = 0.052, 0.38), and negatively associated with thin ideal internalization (Beta = −0.15, 95% CI = −0.23, −0.07) and external eating (Beta = −0.19, 95% CI = −0.30, −0.09). These associations were mainly mediated by BMI. *Conclusions*: Genetic variants associated with BMI are also associated with DE. This association was mediated through BMI suggesting that weight potentially sits on the pathway from genetic liability to DE.

## 1. Introduction

Disordered eating (DE) behaviors [1,2,3], including fasting, binge eating, and related cognitions (such as body dissatisfaction) are widely prevalent in the general population (14–22%) and are considered behavioral and psychological features of a clinical diagnosis of anorexia nervosa (AN), bulimia nervosa (BN), and binge-eating disorder (BED) [1,4,5,6,7]. DE typically arises during pre-adolescence and adolescence [1,7,8] and individuals with DE are at greater risk for mood disorders, psychosocial impairment, and suicidal behavior, as well as at elevated risk for developing a full eating disorder [9,10]. Identification of risk factors that may contribute to DE is an active area of inquiry and may offer new prevention and therapeutic interventions as the current treatment strategies for eating disorders (ED) are limited in their efficacy [11,12].

DE and related cognitions are moderate to highly heritable, with estimates from twin studies ranging between 20–85% making it suitable for genetic analyses [13,14]. Polygenic score (PGS) analyses are an effective method that allow for direct testing of whether common genetic variants associated with one trait are also associated with another [15,16]. PGS approaches to understand DE are of interest as they allow interrogation of the shared genetic etiology of two traits that are associated at the phenotypic level, such as DE and body mass index (BMI) [17,18]. Previous research has highlighted that DE and related cognitions are present across the entire weight spectrum, including AN (at one extreme of the weight spectrum) and BED (often at the other extreme of the weight spectrum) [19]. In addition, children who develop an ED have been shown to follow different BMI trajectories prior to diagnosis [20,21,22]. More specifically, children who go on to develop AN show consistently lower childhood BMI, whereas children who later develop BED show higher premorbid childhood BMIs [20].

It is well established that BMI has a substantial genetic component [23] with the SNP-based heritability ranging between 17–27% [24]. Recent studies have shown a negative genetic correlation between AN and BMI (*r*_g_ ~ −0.24) suggesting shared genetic etiology between AN and BMI, whereby genetic variants associated with higher BMI were associated with lower risk for AN [25,26]. However, less is known about the extent to which DE shares genetic etiology with BMI. 

The first study to investigate the association between DE and a BMI-PGS found that a BMI-PGS is associated with weight loss behaviors [27]. Nagata et al. reported that the BMI-PGS was associated with a higher odds of weight loss behaviors (e.g., dieting, vomiting) and a lower odds of weight gain behaviors (e.g., eating more or different foods than normal). In addition, Nagata et al. found that the association between the BMI-PGS and weight loss behaviors was mediated by measured BMI; a higher BMI-PGS was associated with a higher measured BMI, which in turn was associated with higher odds of engaging in weight loss behaviors. However, this study [27] investigated the association only in a sample of young adults and evidence is currently lacking in adolescence—a critical period for the development of EDs and DE. Furthermore, this study failed to include pre-morbid BMI (measurement of BMI prior to DE) in the mediation analyses, which complicates establishing potential causal pathways that lead from measured BMI to DE. 

In the current study, we investigated whether a BMI-PGS is longitudinally associated with a broad range of DE behaviors and cognitions in a large UK-based population study across adolescence. We hypothesized that the BMI-PGS will be positively correlated with binge eating, emotional eating, and inappropriate compensatory behaviors, such as purging and fasting for weight loss. Based on a previous positive correlation of higher body weight with higher adolescent body dissatisfaction [1] and restrained eating [27,28], we hypothesized that the BMI-PGS will positively correlate with later body dissatisfaction and restrained eating. We also expected a negative association between BMI-PGS and both thin ideal internalization and high external eating—both of which have been found to be negatively correlated with weight or BMI in population-based studies [28,29]. One possible pathway that genetic risk could lead to DE could be through changes in measured BMI which then could lead to increased risk for DE; therefore, we aimed to determine the role of measured BMI as a mediator of these associations. Considering the prospective nature of the data, we also examined developmental differences, determining whether the strength of the association between the BMI-PGS and DE differs across the three time-points during adolescence.

## 2. Methods and Materials

### 2.1. Participants

The Avon Longitudinal Study of Parents and Children (ALSPAC) study is an ongoing population-based birth cohort study of 15,454 mothers and their children (that were born between 1 April 1991 and 31 December 1992) residing in the south west of England (UK) [30,31,32,33]. From the 15,454 pregnancies, 13,988 were alive at 1 year. At age 7 years this sample was bolstered with an additional 913 children. Participants are assessed at regular intervals using clinical interviews, self-report questionnaires, medical records, and physical examinations. We included children based on three waves of data collection which were at age 14 (wave 14, *N* = 10,581), 16 (wave 16, *N* = 9702), and 18 years (wave 18, *N* = 9505). Further details on ALSPAC are available in previous publications [30,32] and the study website contains details of available data through a fully searchable data dictionary: http://www.bristol.ac.uk/alspac/researchers/our-data/. To avoid potential confounding due to relatedness, one sibling per set of multiple births was randomly selected to guarantee independence of participants (*N* = 75). Furthermore, individuals who were closely related to each other, defined as a phi hat > 0.2 (calculated using PLINK v1.90b), were removed; this meant removal of any duplicates or monozygotic twins, first-degree relatives (i.e., parent-offspring and full siblings), and second-degree relatives (i.e., half-siblings, uncles, aunts, grandparents, and double cousins). The authors assert that all procedures contributing to this work comply with the ethical standards of the relevant national and institutional committees on human experimentation and with the Helsinki Declaration of 1975, as revised in 2008.Ethical approval for the study was obtained from the ALSPAC Ethics and Law Committee and the Local Research Ethics Committees (Bristol and Weston Health Authority: E1808 Children of the Nineties: ALSPAC. 28th November 1989. for details see: http://www.bristol.ac.uk/alspac/researchers/research-ethics/). Informed consent for the use of data collected via questionnaires and clinics was obtained from participants following the recommendations of the ALSPAC Ethics and Law Committee at the time. The main caregiver initially provided consent for child participation and from the age 16 years the offspring themselves have provided informed written consent.

### 2.2. Measures

#### 2.2.1. Binary Outcomes

Information on fasting, binge eating, and purging, were assessed at ages 14, 16, and 18 years using questions modified from the Youth Risk Behavior Surveillance System questionnaire [34]. For fasting (N age 14 = 4584, N age 16 = 3844, N age 18 = 2586), participants were asked, “During the past year, how often did you fast (not eat for at least a day) to lose weight or avoid gaining weight?” with the response options “Never”, “Less than once a month”, “1–3 times a month”, “Once a week”, and “2 or more times a week”. This variable was dichotomized as fasting at least once a month in the previous year versus no fasting [8]. For binge eating (N age 14 = 4144, N age 16 = 3336, N age 18 = 1910), participants were asked how often they engaged in overeating (eating a very large amount of food) in the previous year. Participants who answered this question positively were subsequently asked whether they felt out of control during these episodes. We dichotomized the binge eating variable as eating a very large of amount of food at least once a month (with the feeling of loss of control) versus no binge eating. Regarding purging (N age 14 = 4588, N age 16 = 3871, N age 18 = 2582), participants were asked how often they self-induced vomiting or had taken laxatives (or other weight loss medications) to lose weight or avoid weight gain in the previous year; this variable was then dichotomized as purging at least once a month versus no purging.

#### 2.2.2. Continuous Outcomes 

All continuous DE outcomes were assessed at age 14 years and included thin ideal internalization, body dissatisfaction, emotional eating, external eating, and restrained eating. Thin ideal internalization (*N* = 4496) was assessed using the Ideal-Body Stereotype Scale-Revised with gender-specific items used to assess different aspects of appearance-ideal internalization for boys (six items) and girls (five items) [8,35,36]. The responses were rated on a five-point Likert scale from “strongly agree” to “strongly disagree” and the items from this scale were summed to obtain a total score; a higher score corresponded with increased internalization of the thin ideal Body dissatisfaction (*N* = 4625) was assessed using the Body Dissatisfaction Scale [36,37]. Participants were asked gender-specific questions rating their satisfaction with nine body parts with responses on a six-point Likert scale ranging from ‘extremely satisfied’ to ‘extremely dissatisfied’. We constructed a continuous score from this scale with a higher score corresponding to higher body dissatisfaction [8,36]. Restrained eating (*N* = 4530), emotional eating (*N* = 4345), and external eating (*N* = 3995) were assessed using a modified version of the Dutch Eating Behavior Questionnaire (DEBQ; [38])**.** The restrained eating subscale was assessed with two questions, the emotional eating subscale with 14, and the external eating subscale with seven [36,39]. 

#### 2.2.3. Body Mass Index. 

BMI (weight/height^2^) was calculated using objectively measured weight and height obtained during a face-to-face assessment at age 11 years (*N* = 5902) [30,31,32]. Height was measured to the nearest millimeter using a Harpenden Stadiometer (Holtain Ltd., Crymych, UK) and weight was measured using the Tanita Body Fat analyzer (Tanita TBF UK Ltd., Middlesex, UK) to the nearest 50 g. Age- and sex-standardized BMI z-scores (zBMI) were calculated according to UK reference data, indicating the degree to which a child is heavier (>0) or lighter than expected according to his/her age and sex [40].

### 2.3. Genotyping

Genotype data were available for 9915 children out of the total of 15,247 ALSPAC participants. Participants were genome-wide genotyped on the Illumina HumanHap550 quad chip. Details of the quality control checks are described in the Appendix A.

### 2.4. PGS Calculations

PGS were derived from summary statistics of the Genetic Investigation of Anthropometric Traits (GIANT) consortium, referred to as the discovery cohort [41]. The calculation, application, and evaluation of the PGS was carried out with PRSice (2.1.3 beta; github.com/choishingwan/PRSice/) [42,43]. PRSice relies on PLINK to carry out necessary cleaning steps prior to PGS calculation [42,44]. Strand-ambiguous SNPs were removed prior to the scoring. A total of 1,488,001 SNPs were present in both the discovery and in the target cohort. Clumping was applied to extract independent SNPs according to linkage disequilibrium and *p*-value: the SNP with the smallest *p*-value in each 250 kilobase window was retained and all those in linkage disequilibrium (*r*^2^ > 0.1) with this SNP were removed. To calculate the PGS, for each participant the sum of the risk alleles was taken and was then weighted by the effect size estimated from the discovery cohort. The PGS were calculated using the high-resolution scoring (PGS calculated across a large number of *p*-value thresholds) option in PRSice, to identify an optimal *p*-value threshold at which the PGS is optimally associated with the outcome.

### 2.5. Statistical Analyses

#### 2.5.1. Sensitivity Analyses (Linkage Disequilibrium Score Regression) 

The discovery genome-wide association study (GWAS) from which the BMI-PGS was derived included participants from the UK Biobank [41]. Considering a potential overlap in participants between ALSPAC and the UK Biobank we performed a BMI GWAS with biological sex as a covariate using PLINK in the ALSPAC sample [44]. Subsequently we determined the genetic covariance with the BMI GWAS [41] using linkage disequilibrium score regression (LDSC) v1.0.0. [45].

#### 2.5.2. Regression Analyses 

Logistic regressions models were applied for binary outcomes (e.g., fasting at age 14, 16, and 18 years) and linear regression models were used for continuous variables (e.g., thin ideal internalization) with biological sex and the first four ancestry-informative principal components as covariate in all models. We also tested for an interaction between the BMI-PGS and sex and conducted gender stratified analyses. 

As confirmatory analyses, we investigated the association between the BMI-PGS and BMI at age 11 and 18 years and the association between the BMI-PGS and the age and sex standardized BMI (i.e., BMI z-scores) measured at age 11 years. We then investigated the association of DE symptoms at age 14 years, except for purging due to its low endorsement and hence exclusion from our analyses (*N* purging at age 14 years = 74). We repeated the same analyses at age 16 and 18 years for fasting, binge eating, and purging. We report the regression models that explain the largest R^2^ or Nagelkerke’s pseudo-R^2^ at the optimal BMI-PGS *p*-value threshold. For the binary outcome measures we report the Nagelkerke’s pseudo-R^2^ on the liability scale [46]. Empirical *p*-values were calculated through permutation (*N* = 11,000) that account for multiple testing (i.e., number of *p*-value thresholds tested) and overfitting [42]. We additionally calculated False Discovery Rate-corrected *Q*-values adjusting for the number of phenotypes tested [47]. The significance threshold was met if the False Discovery Rate-adjusted *Q* was <0.05.

#### 2.5.3. Generalized Linear Mixed Models (GLMM) 

To understand whether the association of the BMI-PGS with DE differs across the ages (i.e., developmental differences), we used generalized linear mixed models, including BMI-PGS, biological sex, the first four ancestry-informative principal components as fixed effects, and age at self-report as an interaction term. We included random intercepts for each individual in the model, to take into account variance in DE that is due to inter-individual differences. To generate comparable results across the ages, we calculated the PGS at a *p*-value threshold (P_t_) of 1 for each behavioral trait at each age. 

#### 2.5.4. Exploratory Causal Mediation Analysis

The association between BMI-PGS and adolescent DE could be explained through childhood BMI prior to assessment of DE. We used sex- and age-adjusted BMI z-scores (zBMI) measured around the age of 11 years (prior to assessment of DE and related cognitions) to capture a possible pathway linking genetic liability measured by the standardized BMI-PGS to DE (Appendix A). We estimated the BMI-PGS effects that are mediated and not mediated by childhood zBMI scores using concepts proposed in modern causal inference; the natural direct and indirect effects [48]. The natural direct effect (also known as the “average direct effect”, ADE) measures the expected risk difference of the binary outcome measures (e.g., fasting) had the BMI-PGS been hypothetically set to change by 1 SD from 0 to 1, while at the same time childhood zBMI scores had been set to take their natural value (i.e., the zBMI value that would be experienced had the BMI-PGS been set at the reference value of 0, i.e., under no exposure). For the continuous outcome measures (e.g., thin ideal internalization), the ADE measured as the mean difference of the outcome had the BMI-PGS been set to change by 1 SD while childhood zBMI scores were set to take their natural value under no exposure. The natural indirect effect (“average causal mediated effect”, ACME) measures the expected risk difference in binary outcome measures had the BMI-PGS been hypothetically set to take the value 1, while at the same time childhood zBMI scores had been set to take their potential values had BMI-PGS been set to 0 or 1. The ADE was defined in a similar fashion for the continuous outcome measures except that the ACME measured the expected mean difference of the outcome. When summed together, these direct and indirect effects give the total causal effects and therefore are useful measures of the contribution made by a particular pathway to a causal relationship. 

We estimated these effects, expressed as risk differences/mean differences, using the R package ’mediation’ (version 4.4.6; [49]). Our analyses were controlled for biological sex, and the first four ancestry-informative principal components. Furthermore, we assumed that there were no additional unaccounted confounders nor any intermediate confounders [48]. 

#### 2.5.5. Missing Data

Considering the longitudinal nature of ALSPAC, study participants tend to drop out as time goes on leading to missing data. In our analyses we carried out complete cases analyses (CCA) and to counter potential bias introduced by carrying out CCA we included relevant confounders (sex), as well as BMI-PGS, both of which are related to missingness in our data [50]. Hence estimates are unbiased under the missing at random (MAR) assumption.

## 3. Results

### 3.1. Sample Description

Following quality control of the genetic data, a total of 8654 children with genotyping data and at least one outcome measure were included in the analyses (Table 1). Information regarding age, sex, and ancestry for each timepoint can be found in Appendix A. Our sensitivity analyses did not find evidence of sample overlap between our discovery (i.e., BMI GWAS; 25) and target cohort (i.e., ALSPAC) using LDSC, as the intercept from the genetic covariance (0.0068) was less than one standard error (0.0103) from zero, suggesting no sample overlap.

### 3.2. PGS Analyses

#### 3.2.1. BMI

In our confirmatory analyses we found that the BMI-PGS was significantly associated with BMI at age 11 years and 18 years and to zBMI measured at age 11 years (Appendix A).

#### 3.2.2. DE Behaviors 

The BMI-PGS was positively associated with fasting and binge eating at the ages 14, 16, and 18 years and with purging at age 16 years but not at age 18 years (Table 2, Figure 1). There was no significant interaction between the BMI-PGS and sex. When female participants were investigated separately we found similar effect sizes (Appendix A). The effect of the best-fit BMI-PGS on fasting was fairly consistent; for one SD increase in the BMI-PGS children were 1.42 (95% CI 1.25, 1.61), 1.29 (95% CI 1.17, 1.42), and 1.26 (95% CI 1.06, 1.51) times more likely to engage in fasting behavior at age 14, 16, and 18 years, respectively. We did not find evidence for a significant interaction between the PGS-BMI and fasting across the three ages indicating that the effect of the BMI-PGS is not age-dependent. 

Children with higher BMI-PGS were also more likely to report binge eating at all three ages with ORs of 1.28 (95% CI 1.12, 1.46), 1.20 (95% CI 1.08, 1.34), and 1.23 (95% CI: 1.09, 1.39), for the ages 14, 16, and 18 years, respectively. Furthermore, the effect of the BMI-PGS on binge eating did not differ across the three ages. Individuals with higher BMI-PGS were 1.22 (95% CI 1.07, 1.41) times more likely to report purging at age 16 years. We did not find an interaction between the BMI-PGS and age at reporting suggesting that the effect of the BMI-PGS did not differ across the reported ages (Appendix A).

#### 3.2.3. DE Cognitions 

The BMI-PGS was positively correlated with body dissatisfaction and with the DEBQ restrained and emotional eating scale (Table 2). The effect of this association was strongest for body dissatisfaction; children with a higher BMI-PGS (an increase of 1 SD) were more likely to have higher scores on body dissatisfaction at age 14 years (β = 0.99, 95% CI 0.77, 1.22). This effect was less pronounced for the restrained (β = 0.14, 95% CI: 0.10, 1.17) and emotional eating (β = 0.21, 95% CI: 0.05, 0.38) scale of the DEBQ in which a higher BMI-PGS corresponded with lower scores for both behaviors.

In contrast to all other DE outcomes, we found that the BMI-PGS was negatively correlated with thin ideal internalization and with external eating; i.e., individuals with a higher BMI-PGS reported lower scores for thin ideal internalization (β = −0.15, 95% CI: −0.23, −0.07) and external eating (β = −0.19, 95% CI: −0.30, −0.09). There was no significant interaction between the BMI-PGS and sex in relation to the DE cognitions. Furthermore, the effect sizes of the association between the BMI-PGS and DE cognitions did not differ (95% confidence intervals overlapped) when female participants were analyzed separately (Appendix A). 

### 3.3. Exploratory Causal Mediation Analyses

Causal mediation analyses indicated that the zBMI scores mediated the association between the BMI-PGS and DE except for thin ideal internalization (Table 3). Estimate (β_ACME_) of the average causal mediation effect ranged between −0.11 (external eating) and 0.93 (body dissatisfaction). For almost all mediation models, one standard deviation (SD) increase in the BMI-PGS corresponded to an increase in zBMI scores at age 11 years, which in turn led to an increased probability of endorsing DE. The exception to this direction of effect was external eating; one standard deviation increase in the BMI-PGS led to an increase in zBMI scores at age 11 years, which corresponded to lower external eating scores at 14 years. 

## 4. Discussion

To our knowledge this is the largest study to date that has investigated whether a BMI-PGS is longitudinally associated with DE in a general population cohort (N = 8654). We demonstrate that genetic factors that underlie BMI are also associated with DE suggesting possible shared genetic etiology between BMI and DE. This association was mainly mediated through measured age- and sex-adjusted BMI (i.e., zBMI at age 11 years). Our results mirror findings from epidemiological studies that report higher BMI to be positively correlated with binge eating, emotional eating, restrained eating, purging, fasting, and body dissatisfaction, and negatively correlated with external eating [28,51,52,53,54,55]. The negative genetic correlation between the BMI-PGS and thin ideal internalization is also consistent with findings for AN, reported to be negatively genetically correlated (*r*_g_ = −0.32) with BMI [25,26]. The non-significant association between the BMI-PGS and purging at age 18 years was likely due to the relatively rare endorsement of purging behavior (*N* = 166) emphasizing the need for larger samples. Furthermore, we did not find evidence for developmental differences; i.e., the strength of the association between the BMI-PGS and DE did not differ across ages.

It is important to emphasize that our study focused on DE as present in a general population cohort, which has important implications in interpreting our findings. Given the high incidence of overweight and obesity [56] in the general population, more individuals than ever could be at risk of developing DE and particularly those with elevated BMI-PGS could be at greater risk. Consistent with our findings, Nagata et al. reported that higher BMI-PGS is positively correlated with weight loss behaviors (e.g., fasting, use of laxatives) in a population cohort [27]. Findings from our group have also previously implicated BMI-related genes in adolescent binge eating in this sample [51]. Taken together, our data and those of others support the notion of a shared genetic etiology between DE and propensity for higher BMI [27,51,57]. 

Results from our mediation analyses extend previous findings from Reed et al. that reported a causal effect of higher BMI in childhood on increased risk of DE at age 13 years using Mendelian randomization [52]. The importance of BMI in predicting ED has also been highlighted in prospective studies in which BMI trajectories of children who develop an ED during adolescence were found to significantly deviate from children without an eating disorder [20]. We found differences in the estimates of the mediated effect by age- and sex-corrected BMI (zBMI) in the association between the BMI-PGS and DE; suggesting that the effect of actual BMI in late childhood is important for body dissatisfaction, but less for DE behaviors (e.g., purging and fasting for weight loss). As previously shown [1], cognitions such as body dissatisfaction are likely to be influenced by body image distortion and might be more influenced by environmental factors (e.g., comments and teasing about shape and weight). Furthermore, it is important to note that the zBMI scores did not mediate the association between the BMI-PGS and thin ideal internalization. The non-significant mediation for thin ideal internalization might suggest that other pathways apart from prior BMI might account for this association (e.g., environmental factors such as exposure to family or peer factors; [58]). Overall, our findings add to the considerable wealth of literature suggesting an important role for BMI in DE [1,20,25,26,27,52].

This study benefitted from a large discovery sample (*N* ~ 789,224) that included participants from the GIANT consortium and the UK Biobank [41,59] and a relatively large target sample (*N* = 8654). In addition, the availability of a wide range of DE outcomes collected at different time points enriched our analyses. The availability of objectively measured BMI at age 11 years allowed us to investigate the possible role that BMI plays in the association between the BMI-PGS and DE. 

Findings from this study should be interpreted in the context of some limitations. Participants were all recruited from the same geographical region in the southwest of England and therefore the results of this study might not be broadly generalizable to other populations. However, the homogeneity of the sample lends itself to genetic analyses as bias from population stratification is expected to be less pronounced [60]. It is important to note that ED symptoms included in this study were derived from self-reports and the questions asked pertained to the previous year, which may have resulted in misclassification and recall bias. However, we emphasize that observational measures on eating behaviors are not feasible in large cohort studies such as ALSPAC. We did not have information on DE behaviors and related cognitions in late childhood; occurrence of these behaviors prior to the measurement of BMI at age 11 years could have biased estimates of our mediation analysis. However, it is likely that prevalence of these behaviors and cognitions would have been very low in late childhood. We also highlight that the focus of our study was on DE as present in the general population and while this information could improve our understanding of threshold ED, caution should be taken when interpreting these results in the context of threshold ED; a proportion of the individuals endorsing DE do not go on to develop an ED. The dichotomizing of categorical behaviors (fasting, binge eating, and purging) might have resulted in grouping less severe cases with more severe cases and in a loss of variance in the outcome which could have weakened our findings. Furthermore, considering the longitudinal nature of the study, participants tend to drop out as time goes on leading to missing data. Attrition (i.e., loss to follow up) in longitudinal studies such as ALSPAC was reported to be associated with higher BMI-PGS [61]. Given what is known about this cohort, we have assumed that the missingness observed in our data was at random, given certain individual characteristics, including the BMI-PGS; for this reason we do not expect substantial bias affecting our complete cases analyses because they included covariates that were related to missingness in our dataset. We acknowledge that some remaining bias might result from selective attrition (i.e., more severe cases dropping out from the study) [50]. 

The results of our mediation analyses are promising and suggest that joint approaches to the prevention of both obesity and DE might yield better clinical outcomes than those targeting the former or the latter independently [7]. Furthermore, increased awareness amongst pediatricians or general practitioners that children who have overweight or obesity are at increased risk of DE might aid in early screening, detection, and prevention of DE potentially mitigating the development of threshold ED [3]. BMI-PGS can also be used in conjunction with other PGS (e.g., eating disorder PGS) and environmental risk factors in constructing clinical risk prediction models for ED. A robust clinical risk prediction model could for example aid in prediction and guide personalization of treatment, an approach that has shown to be fruitful in many somatic illnesses including coronary artery diseases [62]. However, given the effect sizes observed in this study, the utility of PGS for clinical application for DE is premature. 

In conclusion, our results suggest that genetic propensity for higher BMI is associated with DE, and that this association is mediated by actual measured BMI. Our findings add to the consistent epidemiological literature that implicates BMI in DE. The current study has demonstrated that DE and related cognitions need to be understood in a broad context that includes anthropometry as well as behavioral and developmental components. Future research should focus on multifactorial risk and consider the role of environment as well as genetic predisposition to other DE-related traits. 

## Figures and Tables

**Figure 1 jcm-09-01187-f001:**
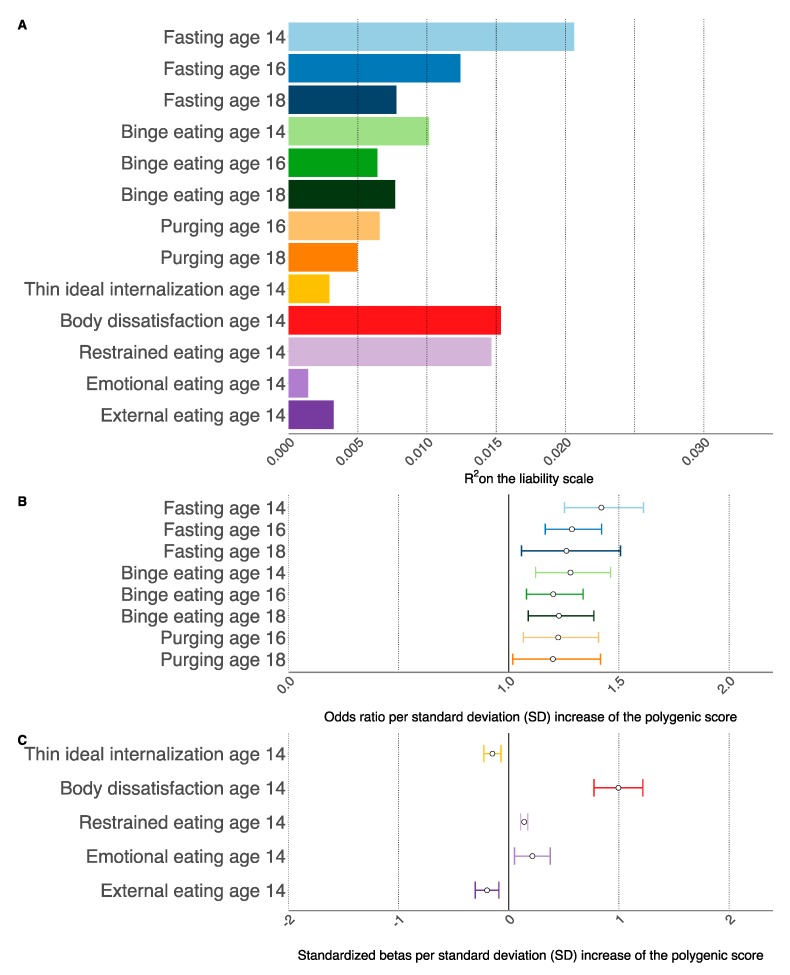
Association between the body mass index polygenic score (BMI-PGS) and disordered eating (DE) behaviors and cognitions in the Avon Longitudinal Study of Parents and Children using generalized linear models [30,31,32,33]. Analyses were corrected for biological sex. Age is measured in years. Sample sizes per outcome were fasting (age 14, n cases = 300, n controls = 4284; age 16, n cases = 516, n controls = 3328; age 18, n cases =143, n controls = 2443;), binge eating (age 14, n cases = 257, n controls = 3887; age 16, n cases = 434, n controls = 2902; age 18, n cases =365, n controls = 1545;), and purging (age 16, n cases = 237, n controls = 3634; age 18, n cases= 166, n controls = 2416), thin ideal internalization (*n* = 4496), body dissatisfaction (*n* = 4625), restrained eating (*n* = 4530), emotional eating (*n* = 4345), external eating (*n* = 3995). (**A**) Explained variance as measured by Nagelkerke’s Pseudo squared multiple correlation (R^2^) for the DE phenotypes. All associations were statistically significant (False Discovery Rate-corrected Q-values < 0.05) except for purging at age 18 years. (**B**) Effect size of the association between the standardized BMI-PGS (to mean zero and standard deviation of one) as measured by odds ratios (ORs) and the DE behaviors. The dots represent the point estimates of the ORs for an increase of one standard deviation in the BMI-PGS and the lines represent the 95% confidence interval of the point estimate. (**C**) Effect size of the association between the standardized BMI-PGS (to mean zero and standard deviation of one) as measured by standardized betas and the DE cognitions. The dots represent the point estimates of the standardized betas for an increase of one standard deviation in the BM-PGS and the lines represent the 95% confidence interval of the point estimate.

**Table 1 jcm-09-01187-t001:** Descriptive statistics of disordered eating behaviors and cognitions and BMI of the participants in the Avon Longitudinal Study of Parents and Children ^a.^

			Cases		Controls	
**Binary Outcomes**	**Age Outcome Measured (Years)**	**Total N**	**N (%)**	**% Female**	**N**	**% Female**
Fasting **^b^**	14	4584	300 (3.4%)	83.7%	4284	53.3%
Fasting **^b^**	16	3844	516 (5.9%)	90.7%	3328	53.8%
Fasting **^b^**	18	2586	143 (1.6%)	93%	2443	62.1%
Binge eating **^b^**	14	4144	257 (2.9%)	71.6%	3887	53.4%
Binge eating **^b^**	16	3336	434 (5.0%)	83.2%	2902	55.3%
Binge eating **^b^**	18	1910	365 (4.2%)	85.2%	1545	62.2%
Purging **^b^**	16	3871	237 (2.7%)	92.4%	3634	56.7%
Purging **^b^**	18	2582	166 (1.9%)	90.9%	2416	62.0%
**Continuous Outcomes**	**Age Outcome Measured (Years)**	**Total N**	**Mean**	**SD**	**Observed Range (min)**	**Observed Range (max)**
Thin ideal internalization **^c^**	14	4496	15.33	2.69	5	25
Body dissatisfaction **^d^**	14	4625	21.85	7.75	9	46.3
Restrained eating **^e^**	14	4530	0.68	1.13	0	5
Emotional eating **^e^**	14	4345	5.22	5.53	0	28
External eating **^e^**	14	3995	8.45	3.34	0	21
Age- and sex-adjusted body mass index (zBMI) ^f^	11	4037	0.59	1.13	−3.22	3.78
**Total sample size ^g^**		8654				

SD = standard deviation. ^a^ Full description of the Avon Longitudinal Study of Parents and Children is described elsewhere; [30,31,32,33]. ^b^ Assessed using questions modified from the Youth Risk Behavior Surveillance System questionnaire [34]. ^c^ Assessed the Ideal-Body Stereotype Scale-Revised with gender-specific items used to assess different aspects for boys and girls [8,35,36]. ^d^ Assessed using the Body Dissatisfaction scale [36,37]. **^e^** As measured by a modified version of the Dutch Eating Behavior Questionnaire [38]. ^f^ Calculated as weight in kilograms divided by height in meters squared. Height was measured to the nearest millimeter using a Harpenden Stadiometer (Holtain Ltd., Crymych, UK) and weight was measured using the Tanita Body Fat analyzer (Tanita TBF UK Ltd., Middlesex, UK) to the nearest 50 g. zBMI was calculated by standardizing BMI by age and sex. ^g^ Individuals with at least one outcome measurement (e.g., fasting, binge eating, body dissatisfaction) and available genotyping data.

**Table 2 jcm-09-01187-t002:** Associations of body mass index polygenic score (BMI-PGS) with disordered eating behaviors and cognitions correcting for biological sex in the Avon Longitudinal Study of Parents and Children.

**Binary Outcomes**	**Age Outcome Measured (Years)**	**Threshold ^a^**	**N SNPs**	***R*^2^**	**OR (95% CI) ^b^**	**Q ^c^**
Fasting	14	0.085	33,379	0.021	1.42 (1.25, 1.61)	<0.001
Fasting	16	0.17	43,956	0.012	1.29 (1.17, 1.42)	<0.001
Fasting	18	0.1	36,088	0.008	1.26 (1.06, 1.51)	0.045
Binge eating	14	0.014	17,929	0.010	1.28 (1.12, 1.46)	0.003
Binge eating	16	0.0016	9523	0.006	1.20 (1.08, 1.34)	0.006
Binge eating	18	0.0047	12,708	0.008	1.23 (1.09, 1.39)	0.008
Purging	16	0.00025	6115	0.007	1.22 (1.07, 1.41)	0.02
Purging	18	0.033	23,731	0.005	1.20 (1.02, 1.42)	0.10
**Continuous Outcomes**	**Age Outcome Measured (Years)**	**Threshold ^a^**	**N SNPs**	***R*^2^**	**β (95% CI) ^d^**	**Q ^c^**
Thin ideal internalization	14	0.5	66,077	0.003	−0.15 (−0.23, −0.07)	0.003
Body dissatisfaction	14	0.0013	9075	0.015	0.99 (0.77, 1.22)	<0.001
Restrained eating	14	0.1	36,088	0.015	0.14 (0.10, 0.17)	<0.001
Emotional eating	14	0.0091	15,467	0.001	0.21 (0.052, 0.38)	0.046
External eating	14	0.0026	10,780	0.003	−0.19 (−0.30, −0.09)	0.003

SNP = single nucleotide polymorphism; R^2^, Nagelkerke’s Pseudo squared multiple correlation on the liability scale for the binary outcome measures and the squared multiple correlation for the continuous outcomes; OR, odds ratio; CI, confidence interval. **^a^** The optimal *p*-value threshold for the inclusion of SNPs in the calculation of the body mass index (BMI) polygenic score (PGS) as determined by PRSice’s high-resolution scoring [42]. **^b^** Odds ratios reflect one standard deviation change in the standardized (to mean zero and standard deviation of one) BMI-PGS. **^c^** Benjamini & Hochberg False Discovery Rate adjustment for the number of phenotypes tested [47]. **^d^** Standardized betas reflect one standard deviation increase in the standardized (to mean zero and standard deviation of one) BMI-PGS.

**Table 3 jcm-09-01187-t003:** Exploratory causal mediation analysis of the association between the disordered eating outcomes and the standardized body mass index polygenic score (BMI-PGS) with the age- and sex-adjusted body mass index z-scores at age 11 years as mediator ^a^. *p*-values for mediation were generated with bootstrapping methods.

Phenotype	Age ^b^	Total Effect Estimate β_total effect_ (95% CI)	Total Effect *p*-Value	Average Direct Effect β_ADE_ (95% CI)	Average Direct Effect *p*-Value	Average Causal Mediation Effect β_ACME_ (95% CI)	Average Causal Mediation Effect *p*-Value
Fasting	14	0.022 (0.01, 0.032)	<0.0001	0.011 (0.001, 0.022)	0.038	0.011 (0.007, 0.014)	<0.0001
Fasting	16	0.035 (0.019, 0.051)	<0.0001	0.021 (0.005, 0.037)	0.004	0.014 (0.008, 0.019)	<0.0001
Fasting	18	0.014 (0.002, 0.028)	0.02	0.009 (−0.002, 0.024)	0.146	0.005 (0.001, 0.01)	0.03
Binge eating	14	0.01 (−0.001, 0.02)	0.07	0.004 (−0.007, 0.014)	0.464	0.006 (0.002, 0.01)	<0.0001
Binge eating	16	0.015 (0.001, 0.031)	0.04	−0.001 (−0.015, 0.014)	0.95	0.016 (0.01, 0.022)	<0.0001
Binge eating	18	0.028 (0.006, 0.053)	0.01	0.01 (−0.013, 0.034)	0.348	0.017 (0.009, 0.026)	<0.0001
Purging	16	0.011 (0.002, 0.022)	0.02	0.007 (−0.003, 0.018)	0.168	0.004 (0.001, 0.008)	0.006
Purging	18	0.014 (0.001, 0.03)	0.04	0.004 (−0.008, 0.019)	0.494	0.009 (0.005, 0.014)	<0.0001
Thin ideal internalization	14	−0.133 (−0.245, −0.017)	0.03	−0.118 (−0.228, −0.001)	0.042	−0.015 (−0.055, 0.027)	0.47
Body dissatisfaction	14	1.097 (0.819, 1.378)	<0.0001	0.171 (−0.097, 0.439)	0.218	0.926 (0.798, 1.061)	<0.0001
Restrained eating	14	0.161 (0.819, 0.208)	<0.0001	0.03 (−0.014, 0.072)	0.162	0.13 (0.113, 0.149)	<0.0001
Emotional eating	14	0.129 (−0.071, 0.356)	0.26	0.03 (−0.181, 0.269)	0.804	0.098 (0.023, 0.176)	0.01
External eating	14	−0.195 (−0.335, −0.053)	0.006	−0.087 (−0.228, 0.06)	0.212	−0.108 (−0.162, −0.057)	<0.0001

^a^ The BMI-PGS was derived from summary statistics of the genome-wide association study (GWAS) carried out by the Genetic Investigation of Anthropometric Traits (GIANT) consortium [41] and were calculated for participants in the Avon Longitudinal Study of Parents and Children [30,31,32,33]. The age- and sex-adjusted BMI (zBMI) at age 11 years was included as a mediator in the causal mediation analyses that was carried out using the R package ´mediation´ (version 4.4.6; 32) which is based on concepts proposed in modern causal inference [48]. Prior to the mediation analyses the BMI-PGS was standardized and the analysis was controlled for biological sex. ^b^ Age was measured in years. ^c^ Measured using the Dutch Eating Behavior Questionnaire (DEBQ; [38]).

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
