# Peer review of "Polygenic Score for Body Mass Index Is Associated with Disordered Eating in a General Population Cohort"

_jcm, 2020, doi:10.3390/jcm9041187_

Round 1
Reviewer 1 Report
This is a manuscript derived from the ALSPAC study, which is an ongoing population-based birth cohort study of 15,454 mothers and their children residing in the south west of England (UK). Body mass index (BMI) polygenic scores were calculated for participants and their association with disordered eating tested. Mediation analyses determined whether the BMI mediated the association between the polygenic risk score of the BMI and disordered eating. The authors found that the BMI-polygenic risk score was positively associated with fasting, binge eating, purging, body dissatisfaction, restrained eating, emotional eating, and negatively associated with thin ideal internalization and external eating.
Overall, this is a well-written manuscript. The introduction is focussed on the topic, but also gives additional background information. The results are well presented. Figure 1 helps the reader to understand the rational behind this research, and Figure 2 and the two tables summarize the results nicely and understandably. The discussion summarized the results and compares them with previous results published in the literature.
As far as I know, I can only agree with the authors that this is the largest study to date that has investigated whether the polygenic risk score for the BMI is longitudinally associated with disordered eating in a general population cohort. This cohort has included more than 8,500 participants.
It could be clinically very relevant to know that genetic factors that underlie BMI are also associated with disordered eating which suggests a possible shared genetic etiology for BMI and disordered eating. However, there is no discussion in the text, which genes and proteins might play a specific role within the overall risk score and whether this role can be targeted by current or future medications. There may also be some implications for psychotherapeutic approaches related to attempts to normalize eating patterns. Even though the whole study is fantastic, and the manuscript is very well written and super-informative for a researcher, a bit more information and discussion about the diagnostic, therapeutic and clinical implications would make this article even more interesting.
Author Response
Comments and Suggestions for Authors
This is a manuscript derived from the ALSPAC study, which is an ongoing population-based birth cohort study of 15,454 mothers and their children residing in the south west of England (UK). Body mass index (BMI) polygenic scores were calculated for participants and their association with disordered eating tested. Mediation analyses determined whether the BMI mediated the association between the polygenic risk score of the BMI and disordered eating. The authors found that the BMI-polygenic risk score was positively associated with fasting, binge eating, purging, body dissatisfaction, restrained eating, emotional eating, and negatively associated with thin ideal internalization and external eating.
Overall, this is a well-written manuscript. The introduction is focussed on the topic, but also gives additional background information. The results are well presented. Figure 1 helps the reader to understand the rational behind this research, and Figure 2 and the two tables summarize the results nicely and understandably. The discussion summarized the results and compares them with previous results published in the literature.
As far as I know, I can only agree with the authors that this is the largest study to date that has investigated whether the polygenic risk score for the BMI is longitudinally associated with disordered eating in a general population cohort. This cohort has included more than 8,500 participants.
Point 1: It could be clinically very relevant to know that genetic factors that underlie BMI are also associated with disordered eating which suggests a possible shared genetic etiology for BMI and disordered eating. However, there is no discussion in the text, which genes and proteins might play a specific role within the overall risk score and whether this role can be targeted by current or future medications. There may also be some implications for psychotherapeutic approaches related to attempts to normalize eating patterns. Even though the whole study is fantastic, and the manuscript is very well written and super-informative for a researcher, a bit more information and discussion about the diagnostic, therapeutic and clinical implications would make this article even more interesting.
Response 1: We thank the reviewer for reviewing our manuscript and for the kind words.
Complex human traits, as investigated in this study, are likely the results of a polygenic genetic architecture; i.e., the combined effect of many genes. In this particular study design, it is not possible to identify specific genes/proteins that may be associated with disordered eating. For this type of information, more fine-grained work using different approaches and designs, such as genome-wide association studies and biological pathway analyses, is required.
The BMI-PGS in our study explained at most 2% of the phenotypic variation in disordered eating. Therefore, we remain cautious regarding our discussion of the diagnostic, therapeutic, and clinical implications which reads:
“The results of our mediation analyses are promising and suggest that joint approaches to the prevention of both obesity and DE might yield better clinical outcomes than those targeting the former or the latter independently [7]. Furthermore, increased awareness amongst pediatricians or general practitioners that children who have overweight or obesity are at increased risk of DE might aid in early screening, detection, and prevention of DE potentially mitigating the development of threshold ED [3]. BMI-PGS can also be used in conjunction with other PGS (e.g., eating disorder PGS) and environmental risk factors in constructing clinical risk prediction models for ED. A robust clinical risk prediction model could for example aid in prediction and guide personalization of treatment, an approach that has shown to be fruitful in many somatic illnesses including coronary artery diseases [62]. However, given the effect sizes observed in this study, the utility of PGS for clinical application for DE is premature.”
Reviewer 2 Report
Dear authors,
Thank you for considering me as a reviewer for this publication in your esteemed journal. I have provided my comments as follows.
Firstly I would like to inform that I don´t have any potential conflict of interest neither any other ethical concerns with regards to the paper:
Polygenic score for body mass index is associated with disordered eating in a general population cohort
In this manuscript authors aimed to research if genetic variants for body mass index were associated with disordered eating. There are plenty of GWAS research in psychiatry in psychosis and bipolar disorder but few in eating disorders and this is, in my view, the main strength of this research. Genome-wide association studies are complicated for clinicians and often far away from the front line of the day-to-day clinic, however I am happy to observe researchers moving on to more complex genetic research like this.
It was a pleasure to review this paper as the authors have shown an expertise and thoroughness that should be appreciated. The authors intend to show whether BMI mediated the association between the BMI-PGS and disordered eating and presents them in a clearly written and well-organized way. The information provided is comprehensive and I like the way it is shown.
The basic analytic model used in this (as in most) GWAS is very simple and considers single genetic markers in isolation. This simple model is not optimal given empirical data as psychiatric disorders are polygenic, and analyses of sets of markers could provide further insight by better reflecting the fundamental genetic architecture. My main concern about this research are indeed common to all GWAS studies in psychiatry, especially in eating disorders: lack of well-defined case and control groups, insufficient sample size, control for multiple testing and control for population stratification…However I think that all these issues and limitations have been taken care of through proper quality control and study setup.
On the other hand and concerning the Avon Longitudinal Study of Parents and Children I would recommend researchers consider how likely non-participation is as a potential source of bias when running genetic association studies and acknowledge this when reporting limitations. E.g. a study of the association between eating disorders and BMI in a selected subsample is likely to be biased by selection, since our genetic results show that both exposure and outcome cause participation; what I mean is that we might lose most serious cases of disordered eating.
In relation to the methodology the most inconvenient issues are:
-The authors don’t specify how was the sample size established neither is too extensive for what these studies require, but this is what it is.
-I don’t like the way the variables were dichotomized; replace “Never”, “Less than once a month”, “1-3 times a month”, “once a week”, and “2 or more times a week” for “at least once a month in the previous year” versus “no fasting” -or whatever it takes- it is a bias that devalues a questionnaire and although I am not able of providing a better way to solve it, it is a limitation that may be included.
I am going to provide some minor proposed changes but, I would insist that the paper is generally well written.
-I would like you to add a reference for the concept of disordered eating (DE) behaviours.
-Authors say that the research has been approved by the Local Research Ethics Committee but they should specify which one and provide an approval number.
- I do not like Figure 1 at all it’s disproportionate, unnecessary and uninformative and I would remove it but it you finally decide to leave it you should provide a title and remove the italics from the description (below).
-I would put the whole table 2 on page 10 alone.
-Table 3 p-value without capital letter
-Figure 2 needs a title and it would be better centered on the page. X-axis in figure 2A and 2B, but above all in 2C are illegible. This colored figure is very similar to Manhattan plots used in this kind of research but it isn’t and might be misleading, initially I didn’t like but on second thought I changed my mind and I appreciate it so let it be.
-In missing data I don’t understand what you mean with attrition; please explain it better.
Please, change this sentence this publication is the work of the authors and MA will serve as…use instead the main author, Mohamed Abdulkadir or M.A (with a point between the initials); this should match the corresponding author.
-In author contributions review JCM submission guides for authors; as far as I am concerned it is not necessary to add the complete name of the authors, just the initials.
I understand the article but, as I am not a native English speaker, I have asked a British member of my team to check the English language and stile and he has told me that moderate English changes required. We have checked the paper for plagiarism by the software provided for our institution and the low returned percentage of similarity would probably indicate that plagiarism has not occurred.
To end I would like to congratulate the authors for the numerous bibliographic references provided and for the great work done.
Yours Sincerely
The reviewer
Author Response
Comments and Suggestions for Authors
Dear authors,
Thank you for considering me as a reviewer for this publication in your esteemed journal. I have provided my comments as follows.
Firstly I would like to inform that I don´t have any potential conflict of interest neither any other ethical concerns with regards to the paper:
Polygenic score for body mass index is associated with disordered eating in a general population cohort
In this manuscript authors aimed to research if genetic variants for body mass index were associated with disordered eating. There are plenty of GWAS research in psychiatry in psychosis and bipolar disorder but few in eating disorders and this is, in my view, the main strength of this research. Genome-wide association studies are complicated for clinicians and often far away from the front line of the day-to-day clinic, however I am happy to observe researchers moving on to more complex genetic research like this.
It was a pleasure to review this paper as the authors have shown an expertise and thoroughness that should be appreciated. The authors intend to show whether BMI mediated the association between the BMI-PGS and disordered eating and presents them in a clearly written and well-organized way. The information provided is comprehensive and I like the way it is shown.
The basic analytic model used in this (as in most) GWAS is very simple and considers single genetic markers in isolation. This simple model is not optimal given empirical data as psychiatric disorders are polygenic, and analyses of sets of markers could provide further insight by better reflecting the fundamental genetic architecture. My main concern about this research are indeed common to all GWAS studies in psychiatry, especially in eating disorders: lack of well-defined case and control groups, insufficient sample size, control for multiple testing and control for population stratification…However I think that all these issues and limitations have been taken care of through proper quality control and study setup.
Point 1: On the other hand and concerning the Avon Longitudinal Study of Parents and Children I would recommend researchers consider how likely non-participation is as a potential source of bias when running genetic association studies and acknowledge this when reporting limitations. E.g. a study of the association between eating disorders and BMI in a selected subsample is likely to be biased by selection, since our genetic results show that both exposure and outcome cause participation; what I mean is that we might lose most serious cases of disordered eating.
Response 1: We thank the reviewer for raising this issue. We acknowledge in the limitation section that, considering the longitudinal nature of the study, participants tend to drop out as time goes on leading to missing data. We agree with the reviewer that a higher BMI PGS are associated with lower participation in the ALSPAC (Taylor et al., 2018). Therefore, we have revised the limitations section with on this point:
“Furthermore, considering the longitudinal nature of the study, participants tend to drop out as time goes on leading to missing data. Attrition (i.e., loss to follow up) in longitudinal studies such as ALSPAC was reported to be associated with higher BMI-PGS [60]. Given what is known about this cohort, we have assumed that the missingness observed in our data was at random (MAR), given certain individual characteristics, including the BMI-PGS; for this reason we do not expect substantial bias affecting our complete cases analyses because they included covariates that were related to missingness in our dataset. We acknowledge that some remaining bias might result from selective attrition (i.e. more severe cases dropping out from the study”
In relation to the methodology the most inconvenient issues are:
Point 2: The authors don’t specify how was the sample size established neither is too extensive for what these studies require, but this is what it is.
Response 2: We have now added more details describing the sample of the study in the methods and materials section:
“The ALSPAC study is an ongoing population-based birth cohort study of 15,454 mothers and their children (that were born between 1 April 1991 and 31 December 1992) residing in the south west of England (UK) [29–32]. From the 15,454 pregnancies, 13,988 were alive at 1 year. At age 7 years this sample was bolstered with an additional 913 children. Participants are assessed at regular intervals using clinical interviews, self-report questionnaires, medical records, and physical examinations. We included children based on three waves of data collection which were at age 14 (wave 14, N = 10,581), 16 (wave 16, N = 9,702), and 18 years (wave 18, N = 9,505). Further details on ALSPAC are available in previous publications [29,31] and the study website contains details of available data through a fully searchable data dictionary: http://www.bristol.ac.uk/alspac/researchers/our-data/.”
Point 3: I don’t like the way the variables were dichotomized; replace “Never”, “Less than once a month”, “1-3 times a month”, “once a week”, and “2 or more times a week” for “at least once a month in the previous year” versus “no fasting” -or whatever it takes- it is a bias that devalues a questionnaire and although I am not able of providing a better way to solve it, it is a limitation that may be included.
Response 3: We acknowledge that dichotomizing the categorical eating behavior problems could have biased our results by grouping more severe cases from the “Once a week” or from the “2 or more times a week” categories with the less severe cases in the “less than once a month” category. However, this was necessary, as some of the more severe categories were endorsed by only a few participants (e.g., N fasting at age 18 “once a week” = 100). Hence, in order to preserve adequate statistical power, we deemed it was the most sensible approach to collapse the categories. We now state this in the limitations section:
“The dichotomizing of categorical behaviors (fasting, binge eating, and purging) might have resulted in grouping less severe cases with more severe cases and in a loss of variance in the outcome which could have weakened our findings.”
I am going to provide some minor proposed changes but, I would insist that the paper is generally well written.
Thank you for that positive comment.
Point 4: I would like you to add a reference for the concept of disordered eating (DE) behaviours.
Response 4: We now reference (Hayes et al. 2018; Micali et al. 2015, 2017) at the first mention of disordered eating; both studies give more information on the concept of disordered eating.
Point 5: Authors say that the research has been approved by the Local Research Ethics Committee but they should specify which one and provide an approval number.
Response 5: We have added a link to the ALSPAC research website which gives more detailed information regarding the local research ethics committees (for details see: http://www.bristol.ac.uk/alspac/researchers/research-ethics/).
Point 6: I do not like Figure 1 at all it’s disproportionate, unnecessary and uninformative and I would remove it but it you finally decide to leave it you should provide a title and remove the italics from the description (below).
Response 6: We acknowledge your concern; however, after careful consideration, we have decided to preserve Figure 1 but to move it to the supplement considering it provides an overview of the different paths of the mediation analyses. To address your recommendations, we have adjusted the proportion of the figure, added a title, and removed the italics from the description.
Point 7: I would put the whole table 2 on page 10 alone.
Response 7: Table 2 in its entirety is now on one page.
Point 8: Table 3 p-value without capital letter
Response 8: p-value in Table 3 is now in lower case.
Point 9: Figure 2 needs a title and it would be better centered on the page. X-axis in figure 2A and 2B, but above all in 2C are illegible. This colored figure is very similar to Manhattan plots used in this kind of research but it isn’t and might be misleading, initially I didn’t like but on second thought I changed my mind and I appreciate it so let it be.
Response 9: The updated title of the figure: “Association between the body mass index polygenic score (BMI-PGS) and disordered eating (DE) behaviors and cognitions in the Avon Longitudinal Study of Parents and Children using generalized linear models”. To make it clearer we have changed the formatting and added the title in bold to separate it from the description text. Furthermore, we centered the figure and increased the font size of the x-axis.
Point 10: In missing data I don’t understand what you mean with attrition; please explain it better.
Response 10: By attrition we mean drop out of the participants from the study.
Point 11: Please, change this sentence this publication is the work of the authors and MA will serve as…use instead the main author, Mohamed Abdulkadir or M.A (with a point between the initials); this should match the corresponding author.
Response 11: We have now corrected this mistake and changed MA to M.A.
Point 12: In author contributions review JCM submission guides for authors; as far as I am concerned it is not necessary to add the complete name of the authors, just the initials.
Response 12: We now refer to the authors by initials.
I understand the article but, as I am not a native English speaker, I have asked a British member of my team to check the English language and stile and he has told me that moderate English changes required. We have checked the paper for plagiarism by the software provided for our institution and the low returned percentage of similarity would probably indicate that plagiarism has not occurred.
Point 13: To end I would like to congratulate the authors for the numerous bibliographic references provided and for the great work done.
Response 13: We would like to thank the reviewer for taking the time to review our manuscript.